# Analysis of *LRRN3*, *MEF2C*, *SLC22A*, and *P2RY12* Gene Expression in the Peripheral Blood of Patients in the Early Stages of Parkinson’s Disease

**DOI:** 10.3390/biomedicines12071391

**Published:** 2024-06-23

**Authors:** Marina V. Shulskaya, Ekaterina I. Semenova, Margarita M. Rudenok, Suzanna A. Partevian, Maria V. Lukashevich, Alexei V. Karabanov, Ekaterina Yu. Fedotova, Sergey N. Illarioshkin, Petr A. Slominsky, Maria I. Shadrina, Anelya Kh. Alieva

**Affiliations:** 1Laboratory of Molecular Genetics of Hereditary Diseases, National Research Center “Kurchatov Institute”, Kurchatova pl., 2, Moscow 123082, Russia; semyonovak@mail.ru (E.I.S.); rudenokmm.img@yandex.ru (M.M.R.); s.partev@yandex.ru (S.A.P.); farouki@mail.ru (M.V.L.); paslominsky@bk.ru (P.A.S.); maria.i.shadrina@yandex.ru (M.I.S.); anelja.a@gmail.com (A.K.A.); 2Federal State Scientific Institution, Scientific Center of Neurology, Russian Academy of Sciences (RAS), Volokolamskoye sh., 80, Moscow 125367, Russia; doctor.karabanov@mail.ru (A.V.K.); ekfed@mail.ru (E.Y.F.); snillario@gmail.com (S.N.I.)

**Keywords:** Parkinson’s disease, gene expression, peripheral blood

## Abstract

Parkinson’s disease (PD) is one of the most common human neurodegenerative diseases. Belated diagnoses of PD and late treatment are caused by its elongated prodromal phase. Thus, searching for new candidate genes participating in the development of the pathological process in the early stages of the disease in patients who have not yet received therapy is relevant. Changes in mRNA and protein levels have been described both in the peripheral blood and in the brain of patients with PD. Thus, analysis of changes in the mRNA expression in peripheral blood is of great importance in studying the early stages of PD. This work aimed to analyze the changes in *MEF2C*, *SLC22A4*, *P2RY12*, and *LRRN3* gene expression in the peripheral blood of patients in the early stages of PD. We found a statistically relevant and PD-specific change in the expression of the *LRRN3* gene, indicating a disruption in the processes of neuronal regeneration and the functioning of synapses. The data obtained during the study indicate that this gene can be considered a potential biomarker of the early stages of PD.

## 1. Introduction

Parkinson’s disease (PD) is one of the most common human neurodegenerative diseases, belonging to the class of multifactorial pathologies. In most cases, this disease is sporadic and is associated with a complex interaction of genetic and environmental factors [1,2]. According to forecasts, the expected number of patients with PD will reach 12 million people worldwide by 2030 [3,4].

The development of PD is primarily associated with the slow and steadily progressive death of dopaminergic neurons in the substantia nigra pars compacta (SN), gradually leading to the appearance of such classical motor symptoms as tremor, rigidity, bradykinesia, and postural instability [5]. Moreover, the manifestation of the first motor symptoms can occur several decades after the onset of the neurodegenerative process [6,7]. A belated diagnosis of PD and late treatment are caused by its elongated prodromal phase. Thus, searching for new candidate genes and pathways participating in the onset and progression of PD in patients who have not yet received therapy continues.

It is known that the CNS is the predominant place where the main pathological processes in PD occur. However, peripheral blood cells are also suitable for the investigation of the processes occurring in the brain in PD. It has been shown that both dopaminergic neurons and blood cells express genes associated with dopaminergic signal transmission [8,9,10,11,12]. Proteins expressed in the peripheral blood may also reflect molecular processes associated with the pathogenesis of PD. Changes in mRNA and protein levels associated with protein degradation [13,14,15,16], mitochondrial dysfunction [14,15,16,17], oxidative stress [16,18], apoptosis [14,15,19,20], and autophagy [21] have been found both in the peripheral blood and in the brains of patients with PD. Thus, analysis of changes in individual mRNA expression in the peripheral blood is potentially relevant for studying the early stages of PD.

This work aimed to analyze changes in *MEF2C*, *SLC22A4*, *P2RY12*, and *LRRN3* gene expression in the peripheral blood of patients in the early stages of PD who received or did not receive therapy. The *P2RY12* gene was selected as a result of a whole-transcriptome study of neuronal progenitor cells and iPSCs from monozygotic twins discordant for PD [22]. The *MEF2C* and *SLC22A4* genes were selected based on the results of analysis of whole-transcriptomic data obtained from mice in a chronic MPTP-induced PD model. The *LRRN3* gene was selected based on the published data [23,24].

## 2. Materials and Methods

### 2.1. Patients

The study was conducted on two cohorts of patients newly diagnosed with PD: those who received and those who did not receive therapy. The diagnoses were made at the Scientific Center of Neurology (Moscow) using the international unified PD rating scale (Unified Parkinson’s Disease Rating Scale, UPDRS) [25] and the Hoehn–Yahr scale [26]. Cohorts of 35 patients with various neurological diseases and 40 healthy volunteers without a history of neurological diseases selected at the Scientific Center of Neurology (Moscow) were used for comparison. The most complete description of the healthy control cohort was given by us earlier in [27]. Both groups of patients with PD are described in detail in Appendix A.

The study was approved by the ethical committee of the Scientific Center of Neurology (Moscow, Russia), protocol 22/5, dated 16 January 2022 [28]. Patients and neurologically healthy volunteers (Russians living in the European part of Russia) were recruited from the Scientific Center of Neurology (Moscow, Russia). All the participants were screened for common PD mutations and had no familial PD. None of the participants had serious comorbidities.

### 2.2. Isolation of Total RNA from Peripheral Blood

Peripheral blood samples were collected at 8 a.m. in the fasting state for subsequent total RNA extraction. Total RNA was isolated from 200 μL of whole blood using the Whole-Blood Total RNA Kit (Zymo Research Corp., Irvine, CA, USA) and the Quick RNA Whole Blood kit (Zymo Research Corp., Irvine, CA, USA) according to the manufacturer’s recommendations. The concentration of isolated total RNA was measured using the Quant-iT RNA BR Assay Kit on a Qubit 3.0 fluorometer (Invitrogen, Carlsbad, CA, USA). Then, yeast tRNA was added [29].

### 2.3. Expression Analysis of Selected Candidate Genes

Analysis of changes in the relative levels of mRNA of the genes was carried out using reverse transcription and real-time PCR with TaqMan probes. The reverse transcription reaction was performed using the RevertAid™ H Minus Reverse Transcriptase kit (Thermo Fisher Scientific, Waltham, MA, USA) according to the manufacturer’s recommendations in a T3 Thermocycler amplifier (T3 Thermoblock, Biometra, Göttingen, Germany). cDNA obtained through a reverse transcription reaction and diluted in a solution of yeast tRNA was used as a template for performing the real-time PCR. Real-time PCR was carried out on a QuantStudio 3 amplifier (Applied Biosystems, Foster City, CA, USA) using PCR reagents (Synthol, Moscow, Russia). To carry out amplification, the following temperature regime was used: 50 °C–60 s, then 40 cycles at 95 °C–15 s and 61 °C–30 s, then 25 °C–30 s. Each sample was analyzed in triplicate.

### 2.4. Data Processing

The sequences of the gene-specific primers and probes for the studied genes are given in Appendix A. The specificity of the primers and probes was checked using BLAST (https://www.ncbi.nlm.nih.gov/tools/primer-blast/, accessed on 22 September 2023) [30,31].

The ΔΔCt amplification threshold comparison method was used for calculation [32]. Statistical processing of the obtained data was carried out using the software package Statistica v. 8.0 (StatSoft Inc., Tulsa, OK, USA) and MS Excel 2019 (Microsoft, Redmond, WA, USA). The data were analyzed using the nonparametric Mann–Whitney U-test.

## 3. Results

The results of the expression analysis in the peripheral blood of patients in the early stages of PD as well as in the neurological control group are presented in Table 1.

As can be seen from Table 1, significant changes in expression in the peripheral blood of patients in the early stages of PD were obtained for three of the four genes studied. However, changes specific to PD were observed only for the *LRRN3* gene since only this gene did not show changes in the neurological control group. A significant decrease in the expression level of the *LRRN3* gene by more than two and three times was shown for both groups of patients with PD (both those who received and those who did not receive therapy), respectively. A considerable and statistically significant increase in expression by more than two times was detected for the *MEF2C* and *SLC22A4* genes in both groups of patients with PD. However, the identified changes are not specific to PD since a similar increase in expression was shown for these genes in the neurological control group.

## 4. Discussion

PD is characterized by a prodromal period of development, and the treatment of patients usually begins immediately upon diagnosis of PD. In this regard, analysis of the gene expression in patients who are in the early stages of PD but have not yet received therapy is a complex and urgent task. Studying changes in gene expression in the peripheral blood allows us not only to search for biomarkers of the early stages of PD but also to study the effect of therapy on gene expression. As can be seen from Table 1, significant changes in expression were obtained in the peripheral blood of patients in the early stages of PD for three of the four studied genes.

The most significant data were obtained for the *LRRN3* gene. This gene encodes leucine repeat-rich neuronal protein 3, which is involved in the development and regeneration of neurons, as well as in the regulation of synaptic connections [33,34]. This gene was isolated as a potential diagnostic biomarker based on the results of whole-transcriptome metadata analysis in two independent studies [23,24]. Our work is the first to show a significant decrease in *LRRN3* expression. It should be noted that a statistically significant decrease in the expression of this gene in patients with PD does not depend on the therapy since similar changes were detected in patients with PD both who received and who did not receive therapy. Moreover, the decrease in the expression of this gene is specific to PD since no statistically significant changes were detected in the neurological control group. Thus, the *LRRN3* gene may be involved in the development of pathological processes in the earliest clinical stages of PD. Apparently, a decrease in the expression of this gene can lead to disruption of the processes of neuronal regeneration and the functioning of synapses [33,34]. In addition, *LRRN3* can be considered a potential biomarker for the earliest stages of PD.

We also detected a significant increase in the expression of the *MEF2C* gene, which encodes a transcription factor involved in the regulation of the functioning of immune cells of myeloid origin, including the microglia, in the formation and differentiation of neurons, and in the growth of axons and dendrites [35,36]. Disruption of the *MEF2C*-PGC1α neuroprotective pathway has been shown to promote mitochondrial dysfunction and lead to apoptotic cell death [35,37,38]. At the same time, an increased level of *MEF2C* expression in humans is positively correlated with improved cognitive function, and in rats, it led to increased differentiation of neuronal progenitor cells into dopaminergic neurons and had a positive effect on motor function [39,40]. The elevated level of *MEF2C*, identified both in the PD group and in the neurological control group, may be a consequence of the development of general compensatory mechanisms aimed at reducing microglial activation, as well as having a positive effect on cognitive and motor functions.

We also found a significant, by more than two times, increase in the expression of the *SLC22A4* gene in all the groups of patients studied. *SLC22A4* is expressed in neurons and neuronal stem cells and encodes the OCTN1 protein [41]. This protein is involved in the transport of various neuroprotective compounds, in particular the antioxidant ergothioneine, thereby helping to reduce oxidative stress [42]. Increased *SLC22A4* expression in both PD patients and neurological controls may be a protective mechanism that alleviates the symptoms of neurological disorders.

*P2RY12* encodes the purinergic receptor P2Y12, which is expressed predominantly by platelets and microglia. Its function has been widely studied in relation to platelet activation and blood coagulation [43], but the role of P2Y12 in neuroinflammation requires further study [44]. Activation of microglial *P2RY12* by ADP/ATP promotes microglial chemotaxis, and increased amounts of these molecules are released by necrotic and apoptotic cells [45]. No significant changes in the expression of *P2RY12* were obtained in our study, which may indicate that it does not participate in the onset and early development of PD and cannot be considered a potential biomarker of this disease.

## 5. Conclusions

Changes in gene expression in the peripheral blood of patients in the early stages of PD, both those who received and those who did not receive therapy, were analyzed in our study. As a result, a statistically significant and PD-specific decrease in the expression of the *LRRN3* gene, which may indicate a disruption in the processes of neuronal regeneration and the functioning of synapses, was shown. The data obtained during the study indicate that this gene can be considered a potential biomarker of the early stages of PD. Apparently, *LRRN3* may be involved in the development of neurodegenerative processes in PD, while the *MEF2C* and *SLC22A4* genes might be involved in the development of general compensatory processes typical for various neurodegenerative diseases (Figure 1).

The limitations of the study include the relatively small size of the analyzed samples, but this limitation is explained by the extremely labor-intensive and time-consuming process of collecting peripheral blood samples from patients in stages 1–2 of PD who have not received therapy.

The findings suggest that *LRRN3* can be considered a potential biomarker only in the early stages of Parkinson’s disease. For other stages of PD pathogenesis, including the preclinical stage and Hoehn–Yahr stages 3–4, additional research is necessary. In addition, the data from this study suggest that *LRRN3* can only be considered a potential biomarker based on its relative mRNA levels using RT-qPCR. Additional studies are also needed to evaluate the corresponding protein as a biomarker for PD.

## Figures and Tables

**Figure 1 biomedicines-12-01391-f001:**
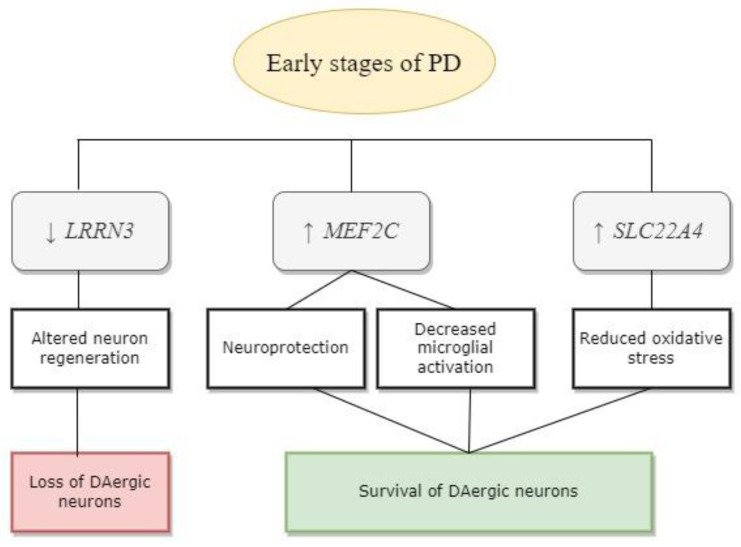
Possible role of the studied genes in PD pathogenesis. The arrows show up- or downregulated gene expression.

**Table 1 biomedicines-12-01391-t001:** Results of analysis of changes in the relative levels of mRNA of *LRRN3*, *MEF2C*, *SLC22A4*, and *P2RY12* genes in the peripheral blood of patients in early stages of PD (fold change relative to healthy control).

Gene	Patients with PD Who Did Not Receive Therapy	Patients with PD Who Received Therapy	Neurological Control
*LRRN3*	**0.32 ^1^ (0.18; 0.51) ^2^**	**0.50 (0.39; 0.56)**	0.91 (0.62; 1.31)
*MEF2C*	**2.56 (2.09; 2.74)**	**2.36 (1.84; 2.93)**	**3.14 (2.48; 3.75)**
*SLC22A4*	**3.26 (2.45; 4.14)**	**2.99 (1.93; 3.33)**	**2.34 (1.50; 3.04)**
*P2RY12*	1.06 (0.85-1.33)	0.59 (0.54; 1.33)	0.91 (0.71; 1.08)

**^1^** Median, **^2^** 25–75 percentiles. Values with *p* < 0.05 are shown in bold. The expression level in the control was set to 1.

## Data Availability

The original contributions presented in the study are included in the article; further inquiries can be directed to the corresponding author. The raw data supporting the conclusions of this article will be made available by the authors on request.

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
