# Peer review of "Analysis of LRRN3, MEF2C, SLC22A, and P2RY12 Gene Expression in the Peripheral Blood of Patients in the Early Stages of Parkinson’s Disease"

_biomedicines, 2024, doi:10.3390/biomedicines12071391_

Round 1

Reviewer 1 Report

Comments and Suggestions for Authors

The article/communication "Analysis of LRRN3, MEF2C, SLC22A, and P2RY12 Gene Expression in the Peripheral Blood of Patients with Early Stages of 3 Parkinson's Disease" does cover an important niche of early-stage PD diagnosis and how certain genes could potentially serve as the disease biomarker in the peripheral blood of the patients. However, there still remain a few concerns that need to be addressed:

·       It’s always recommended to check the levels of the functional protein product of the gene involved in the disease's pathogenesis. Unless a gene is functionally relevant, it cannot be called or promoted as a disease biomarker.

·        Do the authors plan to check the levels of metabolic markers associated with the LRRN3 gene in the samples to substantiate their conclusion further? Some markers of synaptic dysregulation could also be a good target to look at.

·       What are the neurological controls used? Are the mRNA levels in PD patients, with and without therapy, represented relative to this group?

Comments on the Quality of English Language

The English language seems fine. No major grammatical error was detected.

Reviewer 2 Report

Comments and Suggestions for Authors

This work focused on studying changes in the expression of the MEF2C, SLC22A4, P2RY12 and LRRN3 genes in the peripheral blood of patients with early stages of PD.

The study may be interesting to identify the genes involved in PD. However, I would add something else.

For example, I do not see the limitations of the study before the conclusions.

On the other hand, given the sample, the conclusions must be humble.

Likewise, I would add more results, such as average age, percentage by sex and other descriptive results that help understand the sample.

Finally, I would also like to know the results of people with neurological impairment, separated from the healthy ones.

Reviewer 3 Report

Comments and Suggestions for Authors

This is an RT-qPCR study of genes' expressions in blood samples collected from relatively small numbers of persons with 1) early Parkinson's disease (PD) symptoms not on symptomatic medication; 2) early PD and symptomatic medication treatment; 3) subjects with non-PD neurological symptoms; and 4) no neurological disease controls. The authors confirm and earlier study (Guo, et al, 2023) that LRRN3 expression is reduced in both PD groups and provide new data that show that MEF2C expression is also reduced in both PD groups (but not significantly changed in the non-PD neurological control group), and that expressions of both SLC22A4 and P2RY12 were increased in both PD groups, but also were increased in non-PD neurological controls.

This is an interesting paper that portends the expressions in blood of these (and perhaps other genes) as biomarkers of PD. Such a biomarker panel may be useful in clinical diagnosis of early PD, or perhaps even pre-symptomatic PD, that would be most helpful in preventive clinical trials.

I have the following concerns:

1. The authors used non-parametric testing (Mann-Whitney U test) to determine statistical significance. This implies that one of more of their data sets was not normally distributed. Was this the case? If so, which data sets were not normally distributed? 2. The data presentation can be improved. There are several types of graphical plots that would convey information better than the Table used. I would recommend bar plots showing the individual data points, but color coded volcano plots would also work well. Asking the readers to discern bold from normal fonts is too risky a method to show significance. 3. As the authors likely know, the metadata study of Guo, et al (2023) (their reference 23) reported significant reductions in PLOD3 and LNNR3 expressions in blood and brain of PD subjects. Given their findings, why did not the authors also study PLOD3 expression in their samples? This is of concern, as PLOD3 is involved in microglial function, and immune dysfunction is assuming an ever larger role in PD neurodegeneration. Also, there is no compare-and-contrast Discussion of their results compared to those of Guo, et al. (2023)

For these reasons, I suggest that the authors: 1) present their data in a more accessible format and include additional discussion about their statistical methods. They also need to say whether there were any outliers in their data sets (I believe Excel will do this); and 2) compare and contrast their results with those of Guo, et al., 2023; 3) Discuss that their sample sizes were small (particularly since blood, not post-mortem brain, was used) and need to be enlarged.

Round 2

Reviewer 2 Report

Comments and Suggestions for Authors

The authors have made the suggested changes.

Reviewer 3 Report

Comments and Suggestions for Authors

The authors have addressed my prior concerns. Particularly, they have improved their data presentation with box plots. I accept their responses to the other questions I raised.